# Knotwood and Branchwood Polyphenolic Extractives of Silver Fir, Spruce and Douglas Fir and Their Antioxidant, Antifungal and Antibacterial Properties

**DOI:** 10.3390/molecules28176391

**Published:** 2023-09-01

**Authors:** Pauline Gérardin, David Hentges, Philippe Gérardin, Pierre Vinchelin, Stéphane Dumarçay, Coralie Audoin, Christine Gérardin-Charbonnier

**Affiliations:** 1Lermab, Inrae, Faculty of Sciences and Technologies, Université de Lorraine, 54000 Nancy, France; pauline.gerardin@univ-lorraine.fr (P.G.); hentges.david@yahoo.de (D.H.); pierre.vinchelin@outlook.com (P.V.); stephane.dumarcay@univ-lorraine.fr (S.D.); 2Laboratoires Clarins, 95300 Pontoise, France; coralie.audoin@clarins.com

**Keywords:** extract, branch, flavonoid, knot, lignan, structure–activity relationships, valorization

## Abstract

The extractive contents of three softwood species largely used in the wood industry, namely *Abies alba* (Silver fir), *Picea abies* (spruce) and *Pseudotsuga menziesii* (Douglas fir), have been determined quantitatively for knots and at different points chosen along their branches, before analysis using high-performance liquid chromatography coupled with Mass Spectrometry (HPLC-MS). The results indicated that branchwood samples located in close proximity to the stem present high contents of extractives similar to those recorded for the knots. HPLC analysis showed quite similar chemical compositions, indicating that first cm of the branches could be considered as an additional source of knotwood. The antibacterial, antifungal and antioxidant activities of knot’s extractives have been investigated with the dual objective of better understanding the role of high levels of secondary metabolites present in the knot and evaluating their potential for biorefinery applications. The antioxidant activity study showed that crude extracts of Douglas fir knotwood presented higher radical scavenging activity levels than the extracts of Silver fir and spruce, which presented more or less the same activities. Silver fir and spruce knotwood extracts presented higher antibacterial activity levels than the Douglas fir knotwood extracts did, while Douglas fir knotwood extracts presented more fungal growth inhibition than the spruce and fir knotwood extracts did. The structure–activity relationships indicate that radical scavenging and antifungal activities are associated with a higher relative quantity of flavonoids in the crude extracts, while higher relative quantities of lignans are associated with antibacterial activity.

## 1. Introduction

Knots are the parts of the branches that remain embedded in the stem of a tree. Caused by the inclusion of dead or living branches, knots are considered as branch stubs. However, even if knots have been extensively studied, only a few studies have been carried out on the part of the knot included in the branch [1]. The knots of different softwood species have been reported to contain large amounts of secondary metabolites, like lignans [2,3,4,5], which may constitute valuable sources of biomolecules for different applications. The presence of knots in the wood destined for finger jointing or pulp and paper industries is generally considered as problematic, justifying their removal from the wood, thereby allowing the development of a circular economy centered around these types of co-products. However, even if the chemical composition of knotwood has been studied by numerous authors, the reason for the high level of secondary metabolites present in knots of living trees is still unclear.

High levels of extractives in softwood knotwood have been interpreted as a response to stresses that the base of branch is subjected to [6,7]. Branches, and thus knots, develop in reaction to mechanical loads caused by rain, snow and wind, especially in the case of softwoods, which do not lose their leaves during winter. Since lignans have been identified as lignin precursors [8], their amount may be due to higher lignin content around the knots in response to mechanical stress. On the other hand, knots can provide easy access for pathogen attacks, like bacteria or fungi, which may justify higher secondary metabolites contents to avoid the infestation of the stem.

The aim of the present study was first to investigate the content and chemical composition of extractives of branchwood comparatively to those of knotwood. In a second study, the antioxidant, antifungal and antibacterial activities of the three softwood species were investigated to evaluate their biological properties to better understand the role of these secondary metabolites. The structure–bioactivity relationships of knot extracts, but also of pure molecules identified in the crude extracts, were investigated to gain better insights into the role of these secondary metabolites and their possibility of valorization.

## 2. Results and Discussion

### 2.1. Yield of Extracts and Chromatographic Analyses

The yields of extracts obtained from the knots of the three different softwood species, as well as those obtained along the branches after the two successive extractions, are presented in Table 1.

The yields obtained with ethanol are consistent with the results already reported in the literature [4,5]. Fir and spruce contain approximately twice as much of the total extractives contents compared to those of Douglas fir. The evolution of extractives yields of branchwood samples indicated that total extracted yield decreased along the branches to reach higher extractive contents for the disks sampled at close proximity to the stem. As expected, similar to the results recently reported by Vek et al. [1] for Silver fir, the branch wood disk directly sampled at close proximity to the stem contains high levels of total extractives. The extractive contents are quite similar between the knotwood and the first disk of branchwood for Silver fir, while they are lower for the two other species. These contents decreased along the stem to reach values of approximately 2% at the extremity of the branch. The reason for the higher extractive content of the branch wood sampled directly at close proximity to the trunk could be associated with the prolongation of the knot in the branch to assure the good anchoring of the branch to the trunk. These assumptions are supported by the darker color of the center of the disks at close proximity to the trunk. HPLC analyses were carried out to evaluate the nature of the extractives present in knotwood and branchwood. The results of chromatographic analysis are presented in Figure 1.

Comparing the chromatograms of knotwood and branches shows that a part of extractives present in knots were also present at the branch start, especially in the first disk sampled near the main trunk, where the knot are still present. The presence of additional signals other than those present in the knot may be due to the presence of other wood compartments than the knot in the sample disk. The signals of chemical compounds present in the knot decrease as a function of the increase of the distance from the main trunk. Considering the similarities in chemical composition and the quantity of extractives present in the first disk of the branch, the first centimeters of branches could also be a valuable source of the different chemicals present in the knot. Additional investigations carried out in more detail are, however, necessary to evaluate more precisely the potential of such a resource.

The more detailed characterization of the chemical compounds present in knots has been carried out on ethanol extracts in view of their potential valorization. The results of the HPLC-MS analysis are shown in Figure 2 and Table 2. The choice of the positive or negative detection mode by TIC (Total Ion Current), or by UV spectroscopy allowed to improve either the quantity of detected products (TIC being more universal than UV) or the quality of the identification of the different products, with UV giving more structural information than TIC.

Independently of the wood species studied, knot’s ethanol extracts contain mainly high amounts of lignans. Taxifolin, belonging to the flavonoids family, was identified only in Douglas fir knots. The knots of the three softwood species investigated appeared therefore as a valuable source of lignans, which are reported in the literature to possess numerous biological properties [9]. The presence of sugars, sesquiterpenes and resin acids is also indicated by using a mass detector. The nature of each extractive and their relative concentration varied between species. Spruce and fir knots contain principally lignans among which hydroxymatairesinol, secoisolariciresinol, secoisolariciresinol sesquilignan, arctigenin, α-conidendrin, matairesinol, but also oligolignans of higher molecular mass. These results are in agreement with those of Willför et al. [6] who showed presence of many lignans in knots of Picea and Abies species. Smeds et al. [10] also identified oligolignans in knots of spruce, including dimers of hydroxymatairesinol or sesquilignan like oxomatairesinol sesquilignan. Douglas fir knots contain less lignans and more flavonoids among which are taxifolin, pinocembrin and quercetin. The high amounts of taxifolin in Douglas fir was also confirmed by determining the total flavanone content using the Davis method (Figure 3) even if the choice of the standard for the calibration curve, i.e., quercitin in our case, was not the most appropriate. Terpenoids like dehydrojuvabion, todomatuic acid and resin acids were detected in the three softwood species.

### 2.2. Radical Scavenging Activities of the Different Ethanolic Extracts and of Pure Chemicals

Radical scavenging activities are presented in Table 3. Douglas fir extracts present the highest antioxidant activity compared to spruce and fir. Pure quercetin presents the higher antioxidant activity followed by gallic acid, catechin, secoisolariciresinol, taxifolin and HMR. The high concentration of taxifolin in Douglas fir extract probably explains the better inhibition of DPPH radical resulting in a lower IC50. This is so as flavonoids are generally reported to be good antioxidants due to their radical scavenging and chelating properties [11]. Spruce and fir extracts, which contained higher amounts of lignans presenting lower antioxidant activities lead to a higher IC50 value. Secoisolariciresinol, presenting a butanediol structure, was found to present the highest free radical scavenging activity, comparatively to hydroxymatairesinol (HMR). This confirms the results reported by Eklund during his investigation on the structure–antioxidant activity relationship of several lignans [12]. Structure–activity relationships of the different purified or standard compounds tested (Figure 4) confirm the importance of free phenolic groups for their metal-chelating ability and conjugation on radical scavenging activity [12].

Radical scavenging properties are indeed directly associated with the number of free phenolic groups and with the presence of catechol or pyrogallol moieties present in the different compounds. The presence of the carbon–carbon double bond in quercetin allows a strong conjugation with its aromatic B-ring. Thus, hydroxyl radicals with the carbonyl group of the heterocycle C-ring explain the higher antioxidant properties of quercetin. Gallic acid is the second stronger antioxidant, more than catechin and taxifolin. The presence of a methylene group at the C4 of catechin allows the formation of benzylic radicals stabilized by resonance. This occurs during the oxidative process leading to the formation of condensed tannins after dimerization with another catechin radical. It explains the higher antioxidant activity of catechin compared to taxifolin. The carbonyl group at the taxifolin C4 site leads to envisage stabilization of the hydrogen of the C5 phenolic group by the formation of hydrogen bonds, thus explaining its lower antioxidant activity. Silva et al. [13] suggested three structural determinant parameters to be responsible for effective radical scavenging: (i) the catechol B-ring; (ii) the B-ring conjugating with the 4-oxo group via the 2,3-double bond, and (iii) the 3- and 5-OH groups conjugating with the ketone group in C4. These results confirm previous suggestions. Lignans are in general less active antioxidant than flavonoids with the exception of secoisolariciresinol, due to its butanediol moiety [12]. According to Yamauchi et al. [14], the antioxidant activity of lignans is higher for lignans presenting secondary benzylic groups without oxygen. This explains also the higher antioxidant activity measured for secoisolariciresinol when compared to hydroxymatairesinol. From a more general point of view, factors affecting the structure–activity relationship for radical scavenging activities are quite similar for lignans and flavonoids, explaining their approximate similarity of biological activities (Figure 5).

### 2.3. Antibacterial Activities of Ethanol Extracts

Antibacterial activities were present in all knots extracts (Figure 6). The positive control with only *E. coli* presented a strong fluorescence indicating bacterial growth, while samples containing knot extracts showed almost no fluorescence. Fir and spruce presented better antibacterial activities than Douglas fir. Antibacterial activities of fir and spruce extracts remain, however, slightly inferior to that of ethoxyethanol chosen as the representative antibacterial agent. Extracts containing higher amounts of lignans appear to present better antibacterial inhibition effect on Gram-negative bacteria than extracts presenting higher amounts of flavonoids. This is true even if taxifolin, isolated from Acacia Catechu leaf extract, has already been described to possess some antibacterial properties [15].

Effect of the different extracts on the growth of *S. epidermidis*, a Gram-positive bacterium, is reported in Figure 7.

The results showed that an ethanolic extract of Douglas fir and fir knots inhibited the growth of *S. epidermidis*, while that of spruce had a poor inhibition activity against such a bacterium.

Fir knots contain flavonoids such as quercetin and pinocembrin, which are not found in spruce. It would appear that the presence of flavonoids in fir (even in small quantities) and Douglas fir knots enables the inhibition of Gram-positive bacteria. This infers that the antibacterial activity of knot extracts could be dependent on the structure of the cellular wall of the bacterial strain.

Other studies describe investigations on antibacterial activity on flavonoids as an alternative to the increasing resistance of bacteria to classical antibiotics constituting an important health challenge for society [16]. Mixtures of resveratrol, taxifolin, and dihydromyricetin have been reported as good antimicrobial and wound-healing against *Staphylococcus aureus*, *Pseudomonas aeruginosa* and *Candida albicans* [17].

The higher antibacterial effect observed with lignan-rich spruce and fir knot extracts agrees with the literature results describing antibacterial activity of HMR lignin, determined using a disc diffusion method that shows activity against *Staphylococcus epidermidis*, *Candida albicans*, *Proteus* sp. and *Klebsiella* sp. [18].

### 2.4. Antifungal Activities of Ethanol Extracts

The effect of extractives on fungal growth inhibition is presented in Figure 8.

Douglas fir knot extracts presented higher antifungal activities, followed by fir and spruce. Growth inhibition depended of the extract concentrations used for the test, extracts tested at 1000 ppm indicating higher inhibition than those tested at 100 ppm. Knot extracts are more effective against the brown rot fungus *C. puteana* than towards the white rot fungus *C. versicolor*. The inhibition effectiveness was correlated to extract concentration, higher concentration leading to higher inhibition of the fungal growth. Contrary to the results observed for antibacterial activity, extracts containing flavonoids appear more effective to reduce fungal growth than extracts containing more lignans. The effect of flavonoid contents on wood decay durability has already been described in the case of Siberian larch [19,20]. Such a better activity observed for taxifolin may be due to its higher radical scavenging activity. This would allow the slowing down of wood degradation of wood constituents depolymerization by the radicals generated by fungi. The growth inhibitions observed are similar to those reported by Vek et al. [1] on silver fir knot wood extracts where fungal growth inhibition (*T. versicolor*, *S. commune* and *G. trabeum*) was between 15 and 20% for extracts tested at 5% concentration. Lignans extract from knotwood of Norway spruce have also been recently described as potential compounds to fight against grapewine trunk diseases caused by different fungi [21].

According to the different activities measured in our study, it seems that lignans present in high concentration in spruce or fir knots present lower antifungal and radical scavenging activities than flavonoids present in Douglas fir. Conversely, lignan-rich extracts appear to have better antibacterial properties than flavonoid-rich extracts. Even if the chemical composition of ethanolic extracts of the knots of the three softwoods species differs in terms of their nature and of their amount of extractives, it is evident that knots contained higher amounts of bioactive chemical. These may well be involved in the defense of the tree against pathogens. The presence of such high amounts of extractives may be explained by the insertion of the branch into the trunk. This constitutes a fragile point allowing oxygen or pathogens penetration and thus justifying the accumulation of secondary metabolites as a natural defense of the tree. A similar accumulation has also been described in Norway spruce’s natural resistance to needle bladder rust infection [22].

Self-pruning of dead branches is also another parameter which may explain the high amounts of secondary metabolites in knots. Although spruce, silver fir and Douglas fir are known to retain their branches for a long time until self-pruning occurs as a result of density stand or limited luminosity [23,24], the pruning of dead branches exposes the stem wood to different biotic and abiotic degradation agents. It is therefore important that antioxidant, antifungal and antibacterial agents may be present to protect the wounded wood. Moreover, lignans may serve as healing agents to protect the wounded stem from exterior attack in a similar way to what was observed for callus resins. Conifers generally exude oleoresin on being damaged to protect the tree from dehydration and microbial attack. The oleoresin is typically composed of monoterpenes and resin acids. Callus resin is a resin exuded from the callus tissue formed as the wound is closed by the tree’s annual growth. Callus resin is known to contain high amounts of lignans, which are able to cover the surface of dead knots [25]. Further oxidative polymerization of lignans may be responsible for the formation of a surface film to heal and protect stem wood.

Lignans are also considered as intermediates involved in lignin biosynthesis, which implies a random oxidative polymerization of coniferyl alcohol in the case of softwoods. Based on the fact that softwoods are known to form lignin-rich compression wood in response to mechanical stress associated with the load of branches, it can be supposed that the high amounts of lignans detected in softwood’s knots could be associated with higher lignification of the tissues near the knots [4]. However, even if the presence of dilignol, sesquilignol and tetralignol has been observed, the different chemical composition of Douglas fir extract than fir and spruce extracts seems to indicate that polyphenols present in the extracts play other roles in the tree than those associated with their antioxidant, antifungal and antibacterial properties. The rapid decrease of extractive content along the branch suggests also that lignification of reaction wood is not the main reason for the high amounts of lignans detected in knots.

## 3. Materials and Methods

### 3.1. Material

Knotwood: Knots of three economically significant softwoods *Abies alba* (Silver fir), *Picea abies* (spruce) and *Pseudotsuga menziesii* (Douglas fir) were sampled from Savoie Pan company (Tournon, France) and Poirot Construction Bois company (La Bresse, France). Knots were first air-dried before milling at 1100 rpm in a Fritsch Pulverisette 9 (Fritsch, Idar Oberstein, Germany) until no knotwood remained. Knotwood sawdust was oven-dried at 103 °C until constant weight before solvent extraction.Branchwood: Three branches, each from the three softwood species described above, were sampled within the framework of the ExtraFor_Est project (Brennan et al., 2020). The branches were chosen at a height of 1.3 m for each tree. These branches were dried and sliced at different distances from the trunk (0 (near the trunk), 25, 50, 100 and 250 cm) to obtain one 2 cm wide disk for each species. The debarked disks were air-dried and ground at 1100 rpm using a Fritsch Pulverisette 9 (Fritsch, Idar Oberstein, Germany). The sawdust was oven-dried at 103 °C to a constant mass.

### 3.2. Extraction

Two kinds of extraction were performed according to the nature of wood.

Branchwood was Soxhlet-extracted successively with two solvents of different polarity to extract the maximum of extractives presents in branchwood. Extraction was realized using toluene/ethanol mixture (2/1 *v*/*v*) for 7 h followed by ethanol for 16 h on 5 g of dry sawdust using 150 mL of solvent (3 replicates). Solvents were then evaporated under vacuum using a rotary evaporator and the mass of extractives recorded. The percentage of extractives was obtained by dividing the mass of extractives (toluene/ethanol and ethanol) by the dry mass of wood used for extraction, the whole multiplied by 100.

An amount of 5 g of knotwood, containing potential molecules of interest for further valorisation, were Soxhlet-extracted for 6 h using ethanol to avoid utilization of toluene. The extraction was triplicated for each species. After extraction, ethanol was evaporated under vacuum using a rotary evaporator and the extract weighed and stored in a closed vial. The percentage of extractives was calculated as above: $\left[mass\;of\;dry\;extract/mass\;of\;dry\;sawdust\right]\times$100.

### 3.3. Chromatographic Analysis

Dried extracts were resuspended in HPLC grade ethanol at a concentration of 2 mg/mL. The analysis was performed by high-performance liquid chromatography (Nexera Shimadzu chain, Kyoto, Japan) coupled to a triple quadrupole mass spectrometer (LCMS-8030) and a UV-visible diode array detector. The analysis was performed in reverse phase for 23 min. The eluent for the chromatography was composed of a mixture of two solvents: ultrapure water acidified to 0.1% with formic acid and acetonitrile also acidified to 0.1% with formic acid.

### 3.4. Total Flavonoids Content

The total flavonoid content was measured by the Davis method [26] with some modification. Briefly, 200 µL of extract were placed in test tube with 2 mL of diethylene glycol and 200 µL NaOH 1M. Tubes were shaken and incubated at 37 °C for 1 h. The absorbance was measured at 420 nm using a Lambda 365 UV-visible spectrophotometer from Perkin Elmer. Each measurement was realized in triplicate.

### 3.5. Pure Compounds

Purification of lignans 7-hydroxymatairesinol and secoisolariciresinol, and flavonoid taxifolin was carried out by flash chromatography. Briefly, separation was carried out on a C18 grafted silica column with a mixture of water and acetonitrile as eluents using Interchim Puriflash 4100. Other pure compounds such as catechin or quercetin were supplied from Sigma Aldrich (Saint-Quentin-Fallavier, France).

### 3.6. Radical Scavenging Activity

The anti-radical activity of the extracts was measured by UV-visible spectroscopy using the DPPH (2,2-diphenyl 1-picryldhydrazyl) method, which measures the antioxidant capacity of an extract to reduce the purple-colored DPPH radical to yellow-colored DPPH-H. DPPH was prepared at a concentration of 10-4M. The IC50 was calculated using the formula IC50 = (y − b)/a obtained from the equation of the line y = ax + b where a is the directing coefficient and b the y-intercept. Each measurement was realized in triplicate.

### 3.7. Antibacterial Activity

The antibacterial activity of the extracts was measured by a resazurin reduction test, whose blue color becomes fluorescent pink when metabolized by the cells coming from bacterial respiratory chains. The metabolic activity of the cells is measured by reading the fluorescence over time.

Strains of *Escherichia coli* and *Staphylococcus epidermidis* were cultured for 12 h on solid LB medium (Luria Bertani) at 35 °C. The next day, some colonies were taken and placed in a nutrient broth and incubated at 35 °C until an optical density of 0.4 was obtained. The tests were performed in a 96-well microplate in which were introduced the concentrated extracts at 0.5% (m/v) in DMSO, the inoculum, and the resazurin. A control was performed with ethoxyethanol on the extract plate. A positive control was performed to verify the growth of the bacteria in the absence of extracts.

### 3.8. Antifungal Activity

Antifungal tests were performed on *Coriolus versicolor* and *Coniophora puteana*. Extracts of Douglas fir, fir and spruce knots were solubilized in a minimum of ethanol at a concentration of 100, 500 and 1000 ppm. The extracts were then mixed with agar medium and poured into Petri dishes. These plates were then inoculated with or without the fungal mycelium and incubated at 22 °C and 75% relative humidity in a climatic chamber. The growth inhibition was calculated as follows:GI (%) = [(DC − DT)/DC] × 100(1)
where DT is the diameter in cm of the growth zone in the presence of extract and DC the diameter of the growth zone in the control box.

### 3.9. Statistics

The statistical significance of measured differences was analyzed using RStudio software (R4.2.0). The results were analyzed for variance with ANOVA test and the statistically significant *p*-value (0.05) with Tukey’s test.

## 4. Conclusions

The results obtained in this study confirm the potential of knots of different softwood species as a valuable source of polyphenols. Investigations of structure–activity relationships based on antifungal, antibacterial and radical scavenging activities indicate that flavonoids seem to be involved in antifungal and radical scavenging activities, while lignans seem more involved in antibacterial activities against *E. coli*, even if all of these compounds are more or less effective in all the activities tested. Moreover, the chemical diversity of polyphenols present in knot extracts could be an advantage to ensure a broad spectrum of activities. Further investigations will be necessary to gain better insight on the role of the different compounds present in the extracts. Biological activities recorded during this study could be in favor of a protective role of secondary metabolites towards external injuries due to pathogen agents or oxygen during the life of branches or after their self-pruning. The involvement of lignans as intermediates in lignin formation in compression wood seems less obvious due to the rapid decrease of extractive content along the branch. Analysis of extractive content and chemical composition of branch sampled at different distances of the trunk indicated that the samples near the trunk contained higher amounts of extractives. They presented strong analogies with extractives in knot wood, suggesting that the first centimeters of the branch could be an additional source of polyphenols. Biological properties of knot extractives could be of interest for the development of high-value markets like cosmetic, pharmaceutical or nutraceutical applications.

## Figures and Tables

**Figure 1 molecules-28-06391-f001:**
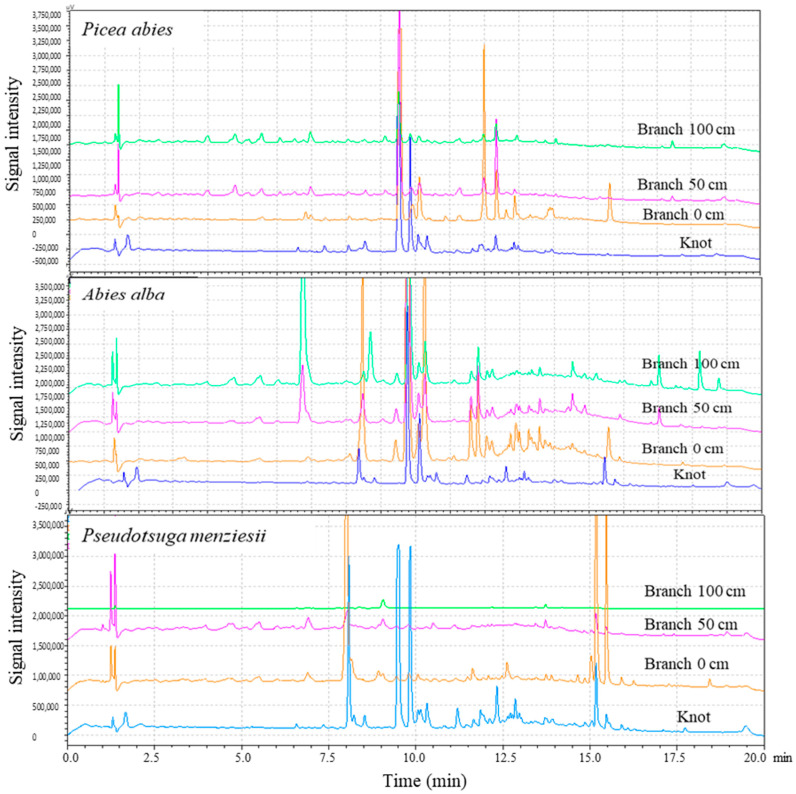
Comparison of UV chromatograms recorded at 190 nm of knotwood and branches extracts of the three softwoods species.

**Figure 2 molecules-28-06391-f002:**
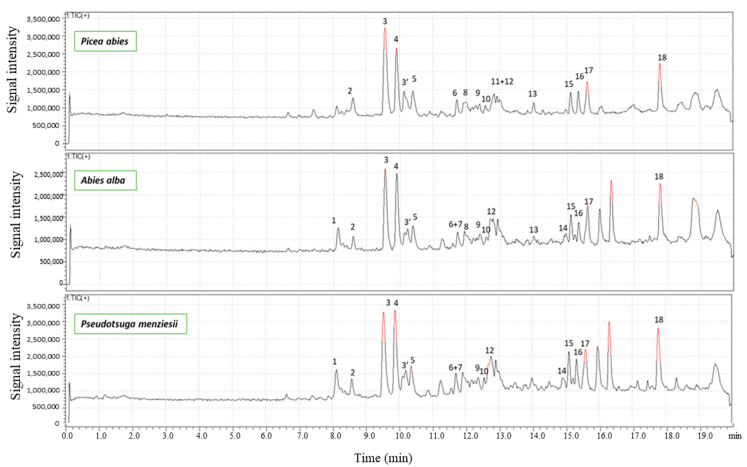
Total Ion Current (+) chromatogram of ethanolic extracts of spruce, fir and Douglas fir knots obtained by HPLC-MS analysis.

**Figure 3 molecules-28-06391-f003:**
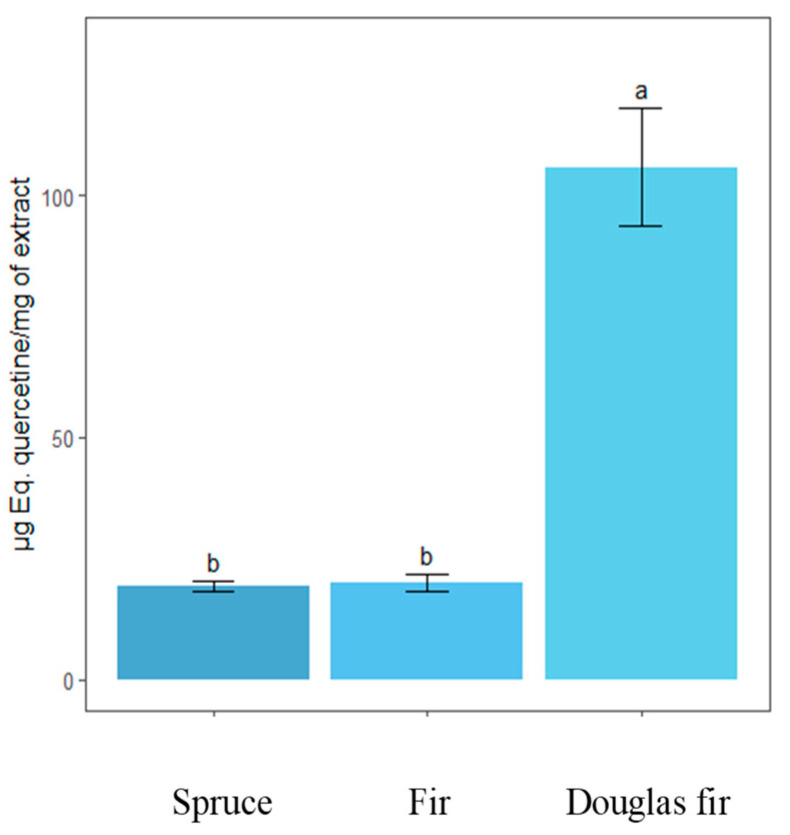
Total flavanone content of knot ethanolic extract of fir, spruce and Douglas fir. A different letter indicates a significantly different result, proven by ANOVA and Tukey statistical tests. The ANOVA gives a *p*-value well below 0.05, indicating that there are differences between the group means.

**Figure 4 molecules-28-06391-f004:**
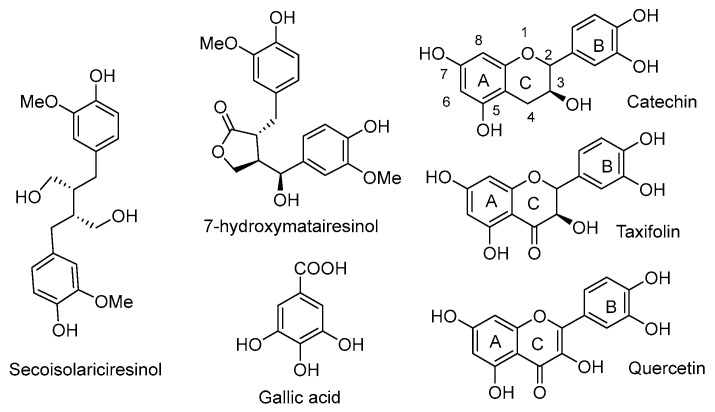
Structure of tested chemical.

**Figure 5 molecules-28-06391-f005:**
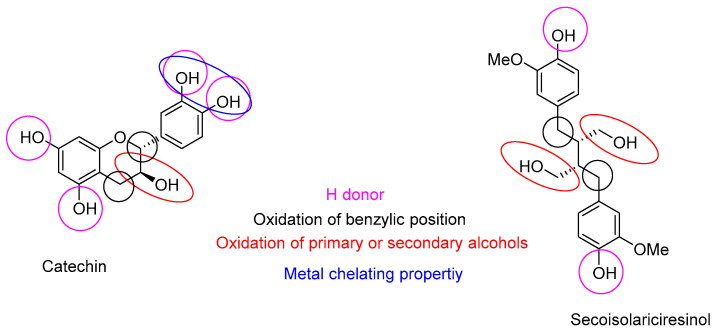
Similarities between flavonoids and lignans for their radical scavenging activities.

**Figure 6 molecules-28-06391-f006:**
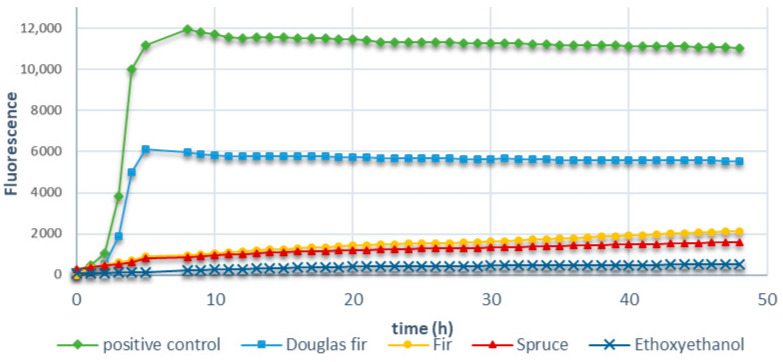
*E. coli* growth in the presence of the different softwood extracts measured using resazurin reduction test.

**Figure 7 molecules-28-06391-f007:**
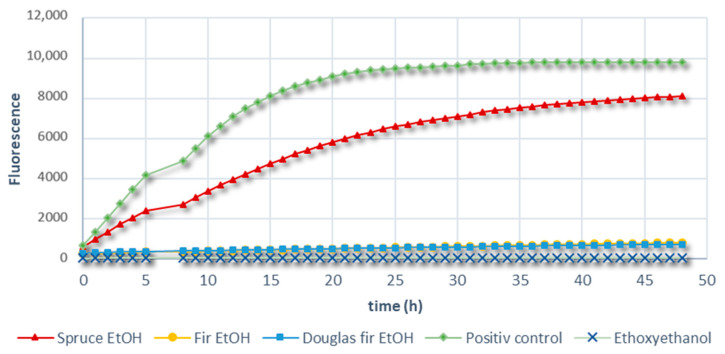
*S. epidermidis* growth in the presence of the different softwood extracts measured using resaeurin reduction test.

**Figure 8 molecules-28-06391-f008:**
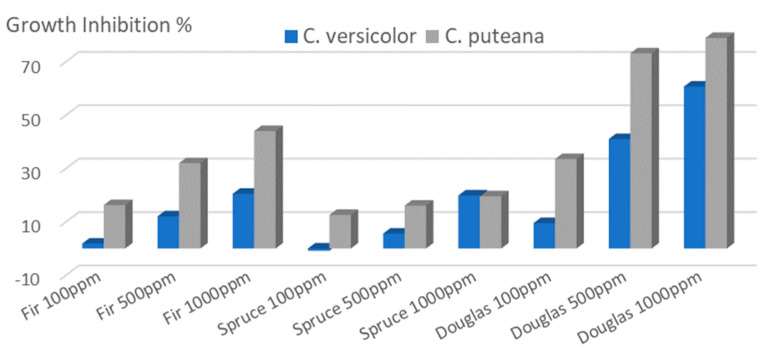
Effect of knot extracts at different concentrations on fungal growth inhibition (after 7 days for *C. versicolor* and 10 days for *C. puteana*).

**Table 1 molecules-28-06391-t001:** Knot extractive contents of the three softwoods species.

Species	Knotwood (%)	Branchwood (%)				
		0 cm	25 cm	50 cm	100 cm	250 cm
*P. abies*	22.69 ± 1.48	14.46 ± 2,10	3.82 ± 0.13	2.41 ± 0.02	1.59 ± 0.52	2.34 ± 1.59
*A. alba*	21.64 ± 0.67	21.59 ± 0.77	11.20 ± 2.42	8.01 ± 1.13	4.11 ± 0.50	1.82 ± 1.01
*P. menziesii*	11.81 ± 1.99	3.80 ± 0.96	1.98 ± 0.46	2.19 ± 0.71	1.85 ± 0.39	2.27 ± 0.97

**Table 2 molecules-28-06391-t002:** Identification of compounds in knots of spruce, fir and Douglas fir by LC-UV-MS.

Compound	Retention Time (min)	Annotation	[M-H]^+^	[M-H]^−^	λ Max	Spuce ^a^	Fir ^a^	Douglas Fir ^a^
**1**	8.61	Conidendric acid	261; 341; 399	375; 421	199; 282	-	++	-
**2**	8.12	Taxifolin	305	303; 607	199; 288	-	-	++
**3**	9.57	Hydroxymatairesinol	327; 397	342; 373; 419	201; 280	+++	++	++
**4**	9.92	Secoisolariciresinol	327; 345; 363; 385	361; 407	196; 280	+++	+++	+++
**5**	10.38	Secoisolariciresinol sesquilignan	493; 581	557	200; 279	++	++	++
**6**	11.72	Arctigenin	359; 399	375; 421	199; 219; 280	++	++	++
**7**	11.72	Quercetin	303	301	199; 219; 280	-	+	+
**8**	11.96	oxomatairesinol	373; 395; 511	371; 551	201; 281	+	+	-
**9**	12.40	α-Conidendrin	357; 398	355; 721	203; 220; 280	++	+	++
**10**	12.57	dimere HMR	587; 770	745	204; 220; 282	+	+	+
**11**	12.83	Dimere lariciresinol-secoisolariciresinol	359; 744	357; 719	202; 220; 281	++	-	-
**12**	12.83	Matairesinol	359	357	202; 220; 281	+	++	++
**13**	14.039	Non identified	373; 413	361; 373; 403	221	+	+	-
**14**	15.17	Pinocembrin	257; 305	255	200; 214; 278	-	++	++
**15**	15.56	Resin acid	219	227; 383; 407	223	++	++	++
**16**	15.96	Resin acid	207; 235; 357; 398	269; 333; 391	222	++	++	++
**17**	16.29	Todomatuic acid	255; 401; 554	253; 321; 389; 529	222	++	++	++
**18**	17.77	Dehydrojuvabion	265	248; 355; 401; 643	224	++	++	++

^a^ (+++) present for the most part in the extract, (++) present in a minority in the extract, (+) present in low quantity in the extract, (-) not detected.

**Table 3 molecules-28-06391-t003:** Radical scavenging activity of different softwood ethanolic extracts and purified or standard compound.

Tested Chemical	IC50 (µg/mL)
Spruce ethanolic extract	54.38 ± 0.01
Fir ethanolic extract	45.81 ± 0.01
Douglas fir ethanolic extract	23.96 ± 0.02
Hydroxymatairesinol	29.43 ± 0.04
Taxifolin	10.06 ± 1.08
Quercetin	1.59 ± 0.02
Secoisolariciresinol	9.39 ± 0.01
Gallic acid	2.04 ± 0.03
Catechin	4.23 ± 0.01

## Data Availability

Data are contained within the article.

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
