# Peer review of "Knotwood and Branchwood Polyphenolic Extractives of Silver Fir, Spruce and Douglas Fir and Their Antioxidant, Antifungal and Antibacterial Properties"

_molecules, 2023, doi:10.3390/molecules28176391_

Round 1

Reviewer 1 Report

The manuscript presents very interesting results on extractives in knot and branch wood. The research design is sound and well described. A few details in methodology are missing. The English needs some editorial corrections. As some help for this I send a few pages a scan. Please search for the word 'present' in the manuscript and try to use synonyms also to improve understanding what you actually mean.

A few detailed remarks:

L12 replace determinate by determined

L23 avoid 3rd person s: .....relationships indicate that.....

L39 change raison to reason

L45 unclear what you mean with 'the latter ones', is it the lignin precursors?

L52 add ....antibacterial activities have been.....

L52 delete hyphen in three softwood species

clause 2 add sample size n for all methods

L64 and 72 it wasn´t saw dust but ground powder

L70 correct to .... 2 cm wide disks.

L71 correct to .....ground at 1100 rpm....

L79 delete successively

L88 replace sawdust by powder

clause 2.3 explain the abbreviation HPLC-MS-MS in detail; use past tense in stead of present tense (was)

clause 2.4 which equipment did you use?

L109 Other pure compounds... and then past tense

clause 2.5 which equipment did you use?

clause 2.6 use past tense in stead of present tense (was)

L127 explain the abbreviation OD

clause 2.7 use past tense in stead of present tense (was)

L134 ....of Douglas fir, fir and spruce knots were.....

clause 2.8 results of statistic analysis are not shown in the results

L158 ...approximately twice of the total extractives content....

L163 ...proximity of the stem...

L165 ...along the branches to reach....

Fig 1 add labels for both axes

L180 .....of knotwood and branches...

L181 ...extractives present in knots....

L184f ....in knot decreased as ......

L191 ...carried out in more detail.....

Fig 2 explain the abbreviation TIC, add labels for both axes

Fig 3 add labels to y axes

L215 is it really proportional? this would be a clearly linear model which is not shown in your graphs, results of statistics are not described.

L266 ....explaining their more....

L285 ...fir knot extracts....

L298 is it really proportional? this would be a clearly linear model, what are your statistics results?

L301 delete sentence because is repeated here from L298

L358 rephrase ....active in all the activities....

nice to read, a few grammar and spelling flaws, the word present is used very very frequently

Author Response

Please see the attachment for answer to reviewer 1

Reviewer 2 Report

The paper presents results regarding the antioxidant, antifungal and antibacterial properties of three different softwood species polyphenolic extractives. The subject is significant for the readers but some observations have to be made:

Page 2, r 67 what species? Same as knots?

Page 2 Extraction  The described methods are standardized? Or existing reference in the literature? Why did you choose those solvents? Why didn’t use the same extraction procedure?

Page 2 Why did you use the Chromatographic analysis? What compounds did you intend to determine? You have to specify in this section, mainly because the preparation method used for samples has to be according to determined analytes.

Page 3 For the total flavonoids content you used a classical semiquantitative spectrophotometric method for measuring total flavanones. Flavanones a re a class of flavonoids and the results are expressed as µg Eq quercetin, which are a flavonol compound. Flavanones and flavonols belong to flavonoids, but the determination method is not specific for all subclasses of flavonoids. I consider that the results reported using the described method are not conclusive.

Page 4 Statistics How many samples did you analyze for forming the data string necessary for statistical interpretation?

Table 1 Use point not comma for results

 Fig 1. What is the wavelength used for chromatograms? Why did you choose this wavelength? How did you establish the relevance of chromatographic conditions for some compounds that you didn’t identify or quantify?

Fig 2 What does it mean “TIC”?

Page 5 r 192 you declared that HPLC-MS-MS was used for the data presented in fig 2 and table 2.

In fig 2 HPLC-MS analysis is declared and in table 2 LC- UV-MS.

At M&M section, LCMS was declared

What is the method used?

Page 7. How can you declare “ The nature of each extractive and their relative concentration varied between species” if you didn’t quantify?

Page 8 r 234  How do you know if the conc of taxifolin is higher or not? You didn’t declare concentrations

Reviewer 3 Report

The research investigates the extractives from the silver fir spruce, and Douglas fir. Silver fir and spruce knotwood extracts presented higher antibacterial activities than Douglas fir knotwood extracts, while Douglas fir knotwood extracts presented higher fungal growth inhibition than spruce and fir knotwood extracts. Structure activity relationships indicates that radical scavenging and antifungal activities are associated to higher relative quantity of flavonoids in the crude extracts, while higher relative quantities of lignans are associated to antibacterial activity. In general, the article is good, but there are some issues that need to be addressed. My suggestions are as follows:

1, The authors should show the main picture showing the extraction of this antimicrobial substance and the pathway to its biomedical use.

2, FT-IR also should be introduced to showing the compound peaks.

3, Some of the antimicrobial and free radical removal mechanisms should be shown in the form of pictures.

4, By the antimicrobial data, what is being shown is still not enough. Staphylococcus aureus also should be detected.

5, Some research should be cited to highlight the potential applications of antibacterials materials. Biomacromolecules 2021, 22, 2, 732–742; Pharmaceutics, 2023, 15(2): 368

Round 2

Reviewer 2 Report

The authors revised their manuscript according suggestions